# Mössbauer Synchrotron and X-ray Studies of Ultrathin YFeO$_3$ Films

Marina Andreeva [1,*], Roman Baulin [1], Aleksandr Nosov [2], Igor Gribov [2], Vladimir Izyurov [2], Oleg Kondratev [3], Ilia Subbotin [3] and Elkhan Pashaev [3]

1   Faculty of Physics, Lomonosov Moscow State University, 119991 Moscow, Russia
2   M.N. Mikheev Institute of Metal Physics RAS, S. Kovalevskaya 18, 620108 Ekaterinburg, Russia
3   National Research Center "Kurchatov Institute", Pl. Kurchatova 1, 123182 Moscow, Russia
*   Correspondence: mandreeva1@yandex.ru

**Abstract:** The YFeO$_3$ orthoferrite is one of the most promising materials for antiferromagnetic (AFM) spintronics. Most studies have dealt with bulk samples, while the thin YFeO$_3$ films possess unusual and variable properties. Ultrathin (3–50 nm) YFeO$_3$ films have been prepared by magnetron sputtering on the *r*-plane (1 $\bar{1}$ 0 2)-oriented Al$_2$O$_3$ substrates (*r*-Al$_2$O$_3$). Their characterization was undertaken by the Mössbauer reflectivity method using a Synchrotron Mössbauer Source and by X-ray diffraction (XRD) including grazing incidence diffraction (GI-XRD). For thin films with different thicknesses, the spin reorientation was detected under the application of the magnetic field of up to 3.5 T. Structural investigations revealed a predominant orthorhombic highly textured YFeO$_3$ phase with (00l) orientation for relatively thick (>10 nm) films. Some inclusions of the Y$_3$Fe$_5$O$_{12}$ garnet (YIG) phase as well as a small amount of the hexagonal YFeO$_3$ phase were detected in the Mössbauer reflectivity spectra and by XRD.

**Keywords:** antiferromagnetic films; Mössbauer spectroscopy; X-ray diffraction





## 1. Introduction

Rare-earth orthoferrites are a family of canted antiferromagnets (AFM) that show an unusual variety of magnetic properties. The canted AFM ordering in ABO$_3$-type orthoferrites provokes the occurrence of ferroelectricity [1–6]. Remarkable properties of orthoferrites such as extremely high domain-wall velocity [7] and the existence of Bloch lines (which separate different magnetization directions inside the domain wall) [8] have significance for applications in magnetic field sensors and magnetooptical data storage devices [9–12]. Using multiferroic materials to construct ever smaller, multifunctional electronic devices is one of the most appealing prospects in the fields of information storage and AFM spintronics.

The yttrium orthoferrite YFeO$_3$ is considered as one of the most promising materials for AFM spintronics alongside other potential technological applications. It has a Néel temperature of ~643 K and exhibits both ferroelectric orderings and weak ferromagnetic behavior at room temperature [3,4]. A large positive magnetocapacitance effect is observed in the YFeO$_3$ single crystal [13]. Nanoparticles of YFeO$_3$ have excellent water photocatalytic properties under visible light irradiation, which are extremely important from the viewpoint of the environmentally friendly utilization of solar energy and developing efficient visible-light active photocatalysts [14,15].

A specific phenomenon of spin reorientations in rare-earth canted AFM has provoked much interest [2,16,17].

Most of the up-to-date studies of YFeO$_3$ compounds have been carried out for bulklike samples, which have been synthetized as nanoparticles, nano-crystallites, or powders by pulsed laser deposition [18], the simple solution method [19], solid state reaction [20,21], the hydrothermal method [14,22,23], sol–gel process [24], the salt-assisted solution combustion

method [12], or mechanosynthesis by high energy ball milling [25,26]. However, modern technologies require thin monocrystalline films whose magnetic properties strongly depend on the film thickness and crystallographic orientation.

The aim of this work was to prepare and investigate monocrystalline ultrathin $YFeO_3$ films by the Mössbauer reflectivity method using a Synchrotron Mössbauer Source (SMS) [27] at the European Synchrotron Radiation Facility (ESRF) [28] and to characterize the magnetic behavior of the $YFeO_3$ films under the applied external field. Additional structural characterization was performed using a Rigaku SmartLab X-ray diffractometer with rotating anode and various diffraction geometries.

## 2. Sample Preparation

Thin films of $YFeO_3$ were grown by *rf* magnetron sputtering. The polycrystalline target was prepared by standard solid state sintering technology from the stoichiometric mixture of the $Fe_2O_3$ and $Y_2O_3$ powders. The final sintering was carried out at 1723 K in air during 5 h. The phase purity of the target was controlled by X-ray diffraction (XRD). Two types of targets were prepared. The first one contained iron with 95% $^{57}Fe$ isotope enrichment. This target was used for the preparation of films for the Mössbauer investigations. The second target was made from the $Fe_2O_3$ powder with a natural content of the $^{57}Fe$ isotope (2.12%). This was used for the preparation of films for the X-ray investigations. Single crystal double-sided epipolished *r*-$Al_2O_3$ (1 $\bar{1}$ 0 2) wafers with the dimensions of 15 × 10 × 0.4 $mm^3$ were used as substrates. The films were deposited using a 90%Ar + 10%$O_2$ gas mixture at the pressure of 0.9 Pa ($9 \times 10^{-3}$ mbar) and discharge power of 100 W. During deposition, the substrate temperature was held at 473 K. Typical deposition rates were about 1.5 nm/min. After deposition, the films were post-annealed in air at 1073 K for 3 h. The thickness of the films was determined using the ZYGO optical profilometer.

For the Mössbauer synchrotron investigations, a series of wedge-shaped $Y^{57}FeO_3$ films were prepared with thicknesses from 3 to 40 nm to cover the wider thickness range. For the films in the thickness range of 3–11 nm, the maximum film thickness gradient over the substrate width did not exceed 0.36 nm/mm, while for the films in the thickness range of 12–40 nm, it did not exceed 1.2 nm/mm.

For the X-ray investigations, a series of $YFeO_3$ thin films with fixed thicknesses in the range from 3 to 50 nm was prepared by the identical technology.

## 3. Mössbauer Reflectivity Spectra

Mössbauer reflectivity spectra (R-spectra) at grazing angles were recorded at the ID18 beamline of ESRF [28] using a Synchrotron Mossbauer Source (SMS) [27] in the temperature ranges of 3.5 K to 273 K and from 273 K to 700 K. To investigate the wedged samples, it was important that the width of the synchrotron beam was only 11 μm and the accuracy of the beam positioning and repositioning did not exceed 100 μm.

The samples were mounted in the cassette holder of the He exchange gas superconducting cryo-magnetic system and the measurements were performed at liquid helium temperature (~3.6 K). An external magnetic field $B^{ext}$ was applied along the beam direction. For the measurement of the X-ray reflectivity curve for 14.4 keV photon energy (0.086 nm), the SMS was detuned from the exact nuclear reflection to the Umweg position, which enlarged the spectrum width from SMS from ~$10^{-8}$ eV (at the pure nuclear Bragg reflection) to ~$10^{-2}$ eV and correspondingly increased the used intensity without distorting the Mössbauer reflectivity. The X-ray reflectivity was measured for the proper adjustment of the zero grazing angle and for the evaluation of the $Y^{57}FeO_3$ film thickness at each point on the wedged sample where the Mössbauer R-spectra were recorded. The angular dependence of the nuclear resonance reflectivity (NRR) was obtained as the integral over the Mössbauer R-spectra at each angle of the incidence.

The reflectivity angular curves and the Mössbauer R-spectra measured for the thickest part of the wedged sample are presented in Figures 1 and 2a, respectively. The Mössbauer R-spectra were measured in the region of the total external reflection (at the grazing angle

of 0.12°, see the vertical mark in Figure 1); in this region, the shape of the R-spectra was similar to the common Mössbauer absorption spectra. For the dominant sextet in the Mössbauer R-spectra in Figure 2, the value of the magnetic hyperfine field is evaluated as $\mathbf{B}_1^{hf} \approx 54.3$ T and undoubtedly belonged to the $^{57}$Fe nuclei in the $Y^{57}FeO_3$ compound [17,29]. Unexpectedly, it occurred that in addition to this sextet, the spectra contained a clearly visible additional sextet corresponding to a smaller magnetic hyperfine field ($\mathbf{B}_i^{hf} \approx 46.5$ T at zero external magnetic field). Variations in the spectra with temperature and under the application of the external magnetic field $\mathbf{B}^{ext}$ make it clear that at least three sextets (as it is shown in Figure 2b) in different proportions could be identified in all spectra, depending upon the thickness of the $Y^{57}FeO_3$ film. The Mössbauer parameters for the two additional sextets ($\mathbf{B}_2^{hf} \approx 46.5$ T, $\mathbf{B}_3^{hf} \approx 53.5$ T) correspond to the ferrimagnetic $Y_3Fe_5O_{12}$ garnet (YIG) [30], in which Fe atoms are located in the two sites (ortho- and tetrahedral) with an occupation of 2:3.

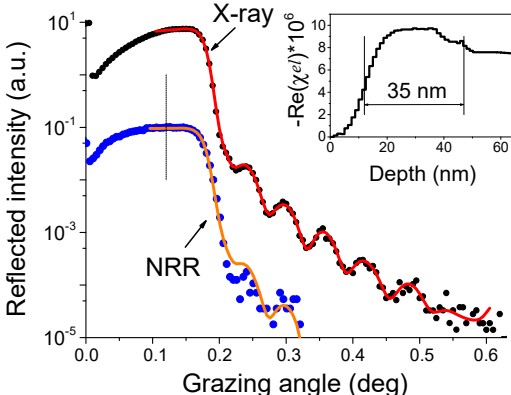

**Figure 1.** The angular dependences of the X-ray and nuclear resonance (NRR) reflectivity near the thickest part of the wedged $Y^{57}FeO_3$ film (23 nm ÷ 40 nm). Curves were normalized and vertically shifted for clarity. Symbols are the experimental data, solid lines are the fit curves. Vertical dash line marks the angle at which the Mössbauer R-spectra were recorded. Insert: the obtained depth profile of the real part of susceptibility (which is approximately proportional to the electronic density). As it follows from this depth profile, the thickness of the $Y^{57}FeO_3$ film can be estimated as ~35 nm.

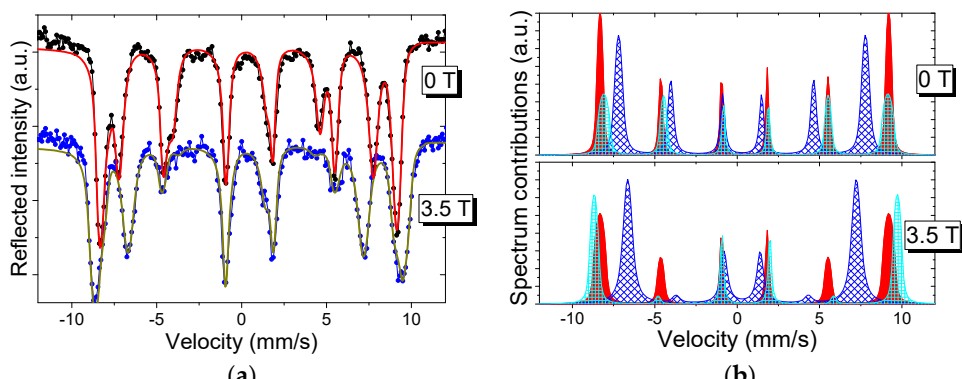

**Figure 2.** (**a**) The Mössbauer R-spectra measured at the grazing angle of 0.12° for the ~35 nm $Y^{57}FeO_3$ film at 3.6 K without the external field (top spectrum) and with a 3.5 T external field applied along the beam (bottom spectrum). The spectra were vertically shifted for clarity. Symbols are the experiment data, lines are the fit curves. (**b**) Fit results: the multiplets contributed to the calculated R-spectra. Dashed sub-spectra correspond to the YIG phase. Spectrum treatment was conducted by our REFSPC program pack [31,32].

The appearance of YIG inclusions in our $YFeO_3$ film was not surprising because, as it had been shown in [11,12,20,31,33], the $YFeO_3$ compound is metastable, while $Y_2O_3$

and $Y_3Fe_5O_{12}$ are thermodynamically stable. The preparation of a single phase yttrium orthoferrite is not simple because of the formation of secondary phases such as $Fe_3O_4$ and $Y_3Fe_5O_{12}$ (see e.g., [18,31,34,35]). Apart from this, the addition of the hexagonal phase of $YFeO_3$ in the orthorhombic matrix can be presented [36]. The Mössbauer data are supported by the X-ray diffraction results (see below).

The remarkable change in the spectrum under the application of the 3.5 T external magnetic field $\mathbf{B}^{ext}$ is shown in Figure 2. The most essential feature is the change in the multiplet splitting. The total field $\mathbf{B}_i^{tot}$ acting on the $^{57}$Fe nuclei is the vector sum $\mathbf{B}_i^{tot} = \mathbf{B}_i^{hf} + \mathbf{B}^{ext}$ and its value changes in accordance with the relative orientations of $\mathbf{B}_i^{hf}$ and $\mathbf{B}^{ext}$. The two sextets belonging to YIG change the splitting in the opposite way: $\mathbf{B}_2^{tot}$ for smaller splitted sextet changes from ~46.5 T to ~43 T. The magnetic moment $\mu_2$ for these Fe atoms directs along $\mathbf{B}^{ext}$, but $\mathbf{B}_2^{hf}$ is antiparallel to $\mu_2$, therefore, $\mathbf{B}_2^{tot}$ decreases on the value of the external field 3.5 T. For the larger splitted YIG sextet, $\mathbf{B}_3^{tot}$ changes from ~53.5 T to ~57 T ($\mu_3$ is antiparallel to $\mathbf{B}^{ext}$ and $\mathbf{B}_3^{hf}$ is parallel to $\mathbf{B}^{ext}$). Therefore, the axis of AFM ordering in the YIG microcrystals is almost completely aligned along the external magnetic field. The disappearance of the 2nd and 5th lines in these sextets confirms their $\mathbf{B}_{2,3}^{hf}$ alignment along the beam direction. Note that for the interpretation of the line ratio in magnetic sextets, it is important to take into account the $\pi$-polarization of the radiation from SMS. Such analysis was presented, for example, in [37].

The sextet belonging to the $Y^{57}FeO_3$ phase practically does not change the splitting but obtains an essential broadening. Therefore, we can conclude that the hyperfine fields $\mathbf{B}_1^{hf}$ at the four $^{57}$Fe nuclei in the $Y^{57}FeO_3$ unit cell obtaine the orientation in the plane perpendicular to the applied field direction (Figure 3b), but not ideally, with slight canting in opposite directions resulting in the line broadening. The orientation of the hyperfine fields $\mathbf{B}_1^{hf}$ at the four $^{57}$Fe nuclei in the $Y^{57}FeO_3$ unit cell without the external field is schematically presented in Figure 3a. Note that there is an uncertainty for the $\mathbf{B}_1^{hf}$ orientation in this case. The picture corresponds to the supposition that $\mathbf{B}_1^{hf}$ is oriented practically in the **a**–**b** plane with slight canting (we chose the longest crystallographic axis as the **c** axis). The ratio of the line intensities in the $Y^{57}FeO_3$ Mössbauer sextet corresponded to the azimuth angle ~29.7° $\pm \Delta$° (relative to the beam direction), where $\Delta$ is the opening angle for the couple of AFM axes in the **a**–**b** plane. Note that the variation of $\Delta$ in the limits 0–30° did not change the line ratio. The slight canting of the AFM alignment (up to ~5–10°) caused a deviation in the $\mathbf{B}_1^{hf}$ vectors from the **a**–**b** plane and the existence of the slight net magnetization along the surface normal for $Y^{57}FeO_3$ also did not change the line ratio in the $Y^{57}FeO_3$ Mössbauer sextet. For the case in (b), the uncertainty in the hyperfine field orientation was removed by the practically unchanged $\mathbf{B}_1^{tot}$ value. The polar angle $\beta$ of the AFM axes with the surface normal was determined as ~27° (Figure 3b).

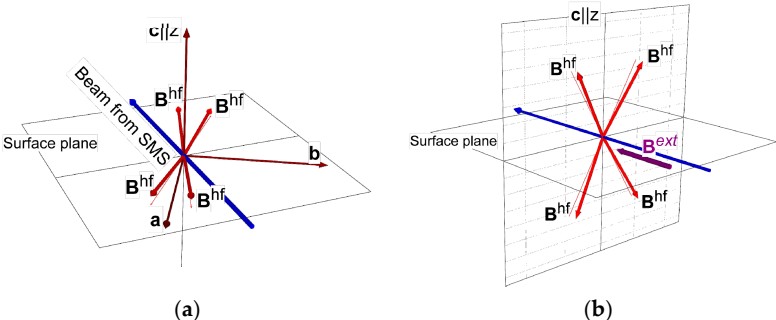

(**a**)          (**b**)

**Figure 3.** The orientation of the hyperfine fields $\mathbf{B}_1^{hf}$ for the four 4 $^{57}$Fe nuclei in the $Y^{57}FeO_3$ unit cell (GxAyFz configuration according to [38,39]) in the geometry of our reflectivity experiment obtained by the fit of the Mössbauer R-spectra in the initial state (**a**) and after application of the 3.5 T external magnetic field (**b**).

The spin reorientation was observed through the measured Mössbauer R-spectra as a function of the $Y^{57}FeO_3$ film thicknesses. The example is shown in Figure 4.

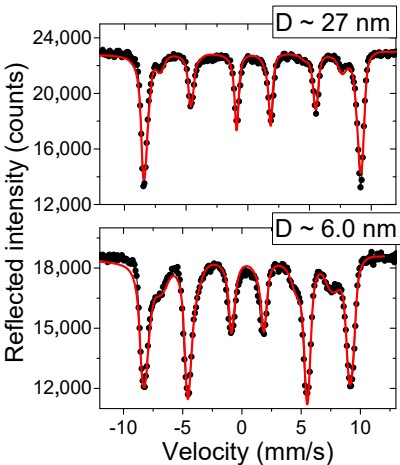

**Figure 4.** The Mössbauer R-spectra of the $Y^{57}FeO_3$ films of the two different thicknesses (~27 nm and ~6 nm) without an external field. Contributions from the YIG phase for these points on the wedged sample were practically negligible.

The line ratio in the magnetic Mössbauer sextet is the visible characteristic of the magnetic hyperfine field orientation, however, with some uncertainty. Suppose that in the total reflection region, the line intensities in the R-spectrum are proportional to the imaginary part of the nuclear resonant scattering amplitude [40] and consequently, neglecting the addition of the waves with the «rotated» polarization to the reflected radiation (for AFM ordering it is always true), the ratio of the line intensities $I_i$ in the magnetic Mössbauer sextet (i = 1, 2, . . . , 6) can be estimated in our grazing incidence geometry for $\pi$-polarized radiation from SMS by the following relation:

$$\frac{I_{2,5}}{I_{1,6}} \cong \frac{4\sin^2\beta \, \cos^2\gamma}{3(\sin^2\gamma + \cos^2\gamma \, \cos^2\beta)} = \frac{4\cos^2\Psi}{3\sin^2\Psi} \tag{1}$$

where $\beta$ is the polar angle for the AFM axis (the direction along which the AFM ordered moments align) relative to the surface normal; $\gamma$ is the azimuth angle relative to the normal to the scattering plane, and $\Psi$ is the angle between the AFM axis and the normal to the scattering plane (that is the direction of the magnetic field of the $\pi$-polarized electromagnetic wave from SMS, which excites the M1 resonant transition in $^{57}Fe$). The designation of the angles is illustrated in Figure 5. From (1), it follows that in the considered case, the line ratio is the same for any orientation of the AFM axis on the cone around the normal to the scattering plane with the apex angle 2$\Psi$. For $\Psi = 0°$, only the 2nd and 5th lines are presented in the Mössbauer R-spectrum, for $\Psi = 90°$, these lines are absent.

The Mössbauer R-spectra measured for the thin and relatively thick $Y^{57}FeO_3$ layers (~27 nm and ~6 nm) shown in Figure 4 clearly demonstrate that the orientation of the AFM axes depends on the film thickness. The intensity of the 2nd and 5th lines in the spectrum for the thinner film was much larger than that for the thicker $Y^{57}FeO_3$ film, which means that in that case, the AFM axes were closer to the normal of the scattering plane. Suppose that for $\mathbf{B}_1^{hf}$ in the **a**–**b** plane for the thicker ~27 nm $Y^{57}FeO_3$ film, the effective angle $\Psi$ for the two AFM axes is fitted as $\Psi \cong 65° \pm \Delta$; for the thinner ~6 nm film, the orientation of $\mathbf{B}_1^{hf}$ is determined by the effective angle $\Psi \cong 48° \pm \Delta$. As it has been determined by X-ray diffraction (see below) for thinner films, the orientation of microcrystals is different from that in the thicker $Y^{57}FeO_3$ films: the crystallographic axis **c** can be oriented for the thinner $Y^{57}FeO_3$ film in the surface plane. Therefore, it can be supposed that the AFM axes for the

thinner ~6 nm film are oriented out of the surface plane (here, we did not take into account the small canting angles that had practically no influence on the line ratio).

Note that a similar spin reorientation was observed for our $Y^{57}FeO_3$ films as the temperature increased [41].

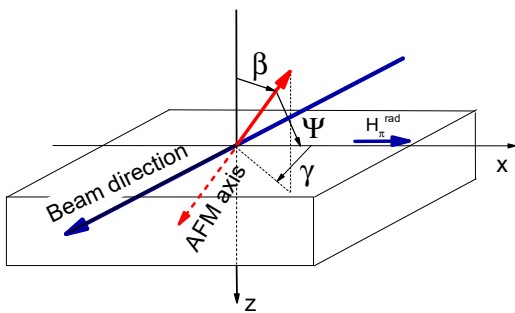

**Figure 5.** The designation of the angles, determining the AFM axis (and $\mathbf{B}_1^{hf}$) orientation in the used geometry. The polarization direction of the magnetic field of the π-polarized electromagnetic wave from SMS corresponds to the x axis. Red thick line (solid and dash) shows the directions of the AFM coupled magnetic moments. For the four Fe atoms in the unit cell of $YFeO_3$, there were two AFM axis mirrors reflected at the **a–c** plane, as shown in Figure 4a. The small canting angles were not taken into account here.

## 4. X-ray Diffraction Study

The structural characterization of ultrathin monocrystalline films is not a simple task (e.g., in [13], the X-ray powder diffraction pattern of the $YFeO_3$ single crystal was obtained from grinding a piece cut from a crack-free single crystal).

In our work, X-ray diffraction investigations of the ultrathin $YFeO_3$ films (thicknesses from 3 nm up to 50 nm) were carried out using the equipment of the Kurchatov complex for synchrotron and neutron investigations of the NRC "Kurchatov Institute" [42]. The measurements were conducted in the two different geometries (Figure 6): standard symmetrical XRD in the θ-2θ scheme (*out-of-plane)* and in the grazing incidence geometry GI-XRD φ-2θ$_φ$ (*in-plane*). Diffraction measurements for the $YFeO_3$ films of different thicknesses (similar to these ones used in our synchrotron Mössbauer experiments) were carried out utilizing a laboratory diffractometer Rigaku SmartLab 9 kW with a rotating Cu anode.

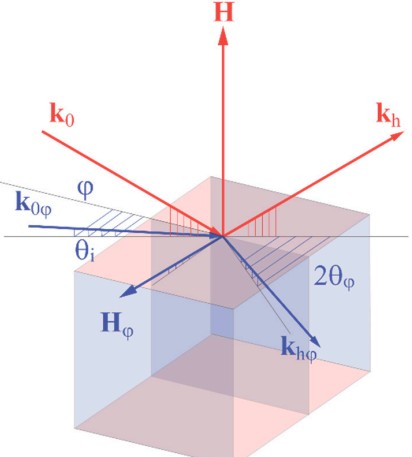

**Figure 6.** Schemes of the *out-of-plane* (red lines) and *in-plane* (blue lines) X-ray diffraction experiments. $\mathbf{k}_0$ and $\mathbf{k}_h$ are vectors of the incident waves and diffracted from planes that are parallel to the sample surface, and **H** is the diffraction vector for the *out-of-plane* geometry. $\mathbf{k}_{0φ}$ and $\mathbf{k}_{hφ}$ are vectors of the incident waves and diffracted from planes that are perpendicular to the sample surface, and $\mathbf{H}_φ$ is the diffraction vector for the *in-plane* geometry.

The measurements in the *out-of-plane* geometry were conducted using the high-resolution scheme, which includes a parallel beam mirror, precise crystal monochromator 2x-Ge(220), and collimating slits placed before the sample and detector. The divergence of the beam incident at the sample did not exceed $0.1°$. The X-ray diffraction curves were measured in a wide-angle range in a step $\theta$-$2\theta$ mode. The diffraction scheme was preliminarily aligned to a precise Bragg condition for the $(\bar{1}\,0\,1\,2)$ reflection of the sapphire substrate.

The measurements in the *in-plane* geometry were conducted in the scheme utilizing a parallel beam mirror, Soller slits with an angle acceptance of $2.5°$ placed before the sample and detector, and the CuK$_\beta$-filter. The angle of incidence $\theta_i$ of the X-rays on the studied sample $\theta_i$ was chosen close to the critical angle, which ensured that the penetration depth of the X-ray radiation did not exceed the thickness of the studied films. Under these conditions, the wave field was magnified by the factor of ~4, which resulted in more intense diffraction patterns from the thin and ultrathin films. The disadvantage of such a scheme is the low spatial resolution. The X-ray diffraction curves were measured in a step $\varphi$-$2\theta_\varphi$ mode (e.g., XRD in a wide-angle range) [43,44]. The diffraction vector for the GI-XRD scheme lies in the horizontal plane, while for the XRD scheme, it is in the vertical plane. The X-ray diffraction curves in the GI-XRD scheme were measured by changing the azimuthal angle $\varphi$ while simultaneously rotating the detector in the horizontal plane at the angle $2\theta_\varphi$, keeping unchanged the orientation of the scattering vector for the chosen YFeO$_3$ reflection.

The analysis of the measured diffraction $\theta$-$2\theta$ curves was performed by using CIF (Crystallographic Interchange File) files for orthorhombic YFeO$_3$ [45], hexagonal YFeO$_3$ [20], and Al$_2$O$_3$ [46]. For confirmation of the results, the X-ray powder diffraction patterns of these compounds as well as the theoretical diffraction curves for the epitaxial film/substrate case were calculated using the program VESTA (Visualization for Electronic Structural Analysis) [47,48] and the calculation algorithm from [49]. These theoretical dependencies are presented in the Supplementary Materials. Note that in this paper, the identification of reflections (h k l) for orthorhombic YFeO$_3$ was adopted for the case **a** < **b** < **c**, where **a**, **b**, **c** are the parameters of the crystal lattice.

The measured diffraction $\theta$-$2\theta$ curves for the samples with the film thicknesses of 4 nm and 40 nm are presented in Figure 7.

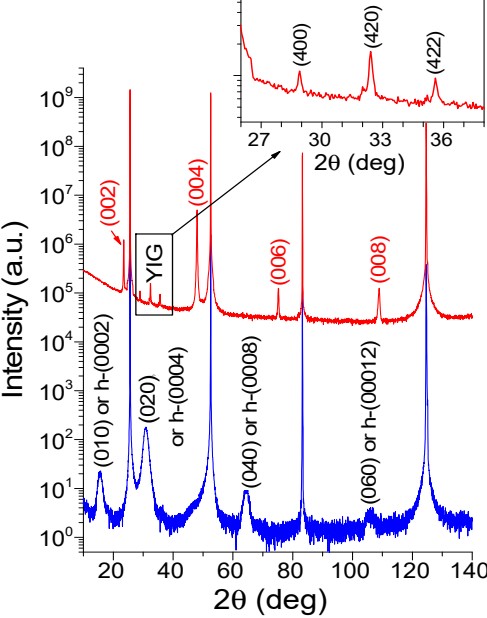

**Figure 7.** The diffraction $\theta$-$2\theta$ curves for the YFeO$_3$ film with the thickness of 4 nm (bottom curve) and 40 nm (top curve). The peaks from YFeO$_3$ are labeled. The "h"-symbol marks the possible reflections from the hexagonal YFeO$_3$ phase. The diffraction peaks from the YIG phase are observable for the 40 nm film in the angular range ~26–37°, as it is shown in the insert.

The narrow and intense peaks originated from the *r*-$Al_2O_3$ substrate. The sequence of diffraction peaks from the $YFeO_3$ phase was labeled. The analysis of the diffraction patterns revealed that for the sample with the film thickness of 40 nm, the observed peaks corresponded to (0 0 l) reflections from the orthorhombic $YFeO_3$. The presence of peaks from the strictly one type of crystal planes indicates a well-formed [0 0 l] texture of the film. Orthoferrites of the general formula $RFeO_3$ have the orthorhombic structure (Pnma/Pbnm space group), which can be derived from the ideal perovskite structure by rotations (inclinations) of its $FeO_6$ octahedra. Each of the four $Fe^{3+}$ ions in the crystal unit cell is surrounded by six oxygen atoms forming an octahedron. The crystal unit cell parameters for $YFeO_3$ are determined from the XRD pattern as follows: **a** = 0.527 nm, **b** = 0.558 nm, **c** = 0.759 nm, these values correspond to the literature data [4] and international database [50]. Note that the notation convention for the **a**, **b**, **c** for orthoferrites and respectively the indices in the diffraction patterns are different in some older papers (see e.g., [2,13,15,19,51–53]), namely the largest lattice parameter is **b**.

The diffraction curves for the 40 nm film also include the low-intensity maxima in the angular range ~26–37°, which corresponded to the reflections from the YIG, which are clearly seen in the insert in Figure 7.

The diffraction pattern for the ultrathin film of 4 nm is different from that for the 40 nm film. It also contains narrow peaks from the *r*-$Al_2O_3$, but the most essential difference is that the wide maxima from the $YFeO_3$ film now correspond to the (0 k 0) reflections of the orthorhombic $YFeO_3$. This means that for this thickness, the crystallographic axis c now lies in the surface plane. At the same time, it is known that the hexagonal crystallographic modification of $YFeO_3$ exists with lattice parameters **a** = **b** = 0.607 nm and **c** = 1.173 nm (P6/3cm space group) [36,54,55] and you can make sure that the (0 0 0 l) reflections from the hexagonal $YFeO_3$ correspond to the (0 k 0) reflections from the orthorhombic $YFeO_3$.

The GI-XRD measurements were performed at different azimuth orientations of the investigated films. As illustrated by Figure 8, we managed to detect some diffraction peaks in this grazing incidence geometry. The analysis of their angular positions for the sample with the film thickness of 40 nm supports the assumption of the sharp [0 0 l] texture of the $YFeO_3$ film by the presence of reflections from the (0 1 0), (1 1 0), (2 1 0), and (1 0 0) planes. The GI-XRD patterns for the ultra-thin 4 nm film differ substantially. The observed diffraction maxima correspond to reflections from both the (0 0 1), (1 0 0) planes of orthorhombic $YFeO_3$ and the (1 1 $\bar{2}$ 0), (1 0 $\bar{1}$ 0) planes of the hexagonal $YFeO_3$ (marked by "h"-symbol in Figure 8). This means that the orientation of the microcrystals in the ultrathin film is different from that for the thicker film. Thereby, the 40 nm film has the vivid texture of [0 0 l] with the largest side of the $YFeO_3$ unit cell oriented normally to the substrate surface (with the small insertions of YIG). The 4 nm film, presumably, has been formed as an "island-like" film in which the orthorhombic $YFeO_3$ has the largest side of the unit cell oriented parallel to the surface. There are also some islands of the hexagonal $YFeO_3$ with the (0 0 0 l) orientation. The small number of islands of YIG is also observable by the presence of the corresponding diffraction peaks. Its absence in the XRD patterns for the 4 nm film recorded in the *out-of-plane* geometry can be explained by the small quantity of YIG reflecting centers for such a small thickness of the film.

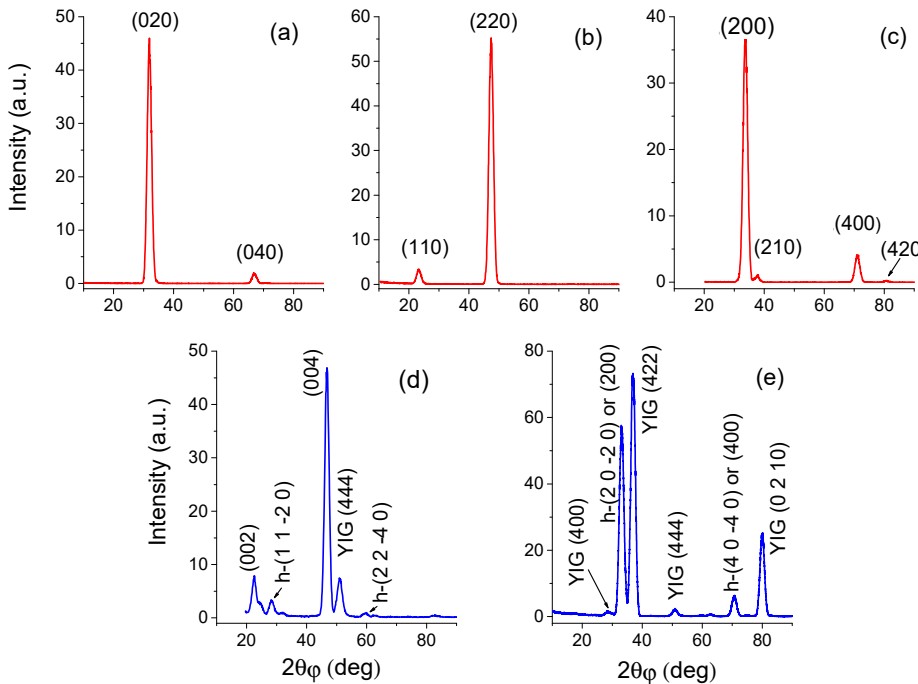

**Figure 8.** The GI-XRD patterns for the 40 nm YFeO$_3$ film (graphs (**a**–**c**)) and for the 4 nm YFeO$_3$ film (graphs (**d**,**e**)). The chosen azimuth orientations of the scattering vectors for the 40 nm YFeO$_3$ film were (0 k 0) (**a**), (h k 0) (**b**), and (h 0 0) (**c**). The choice of the azimuth orientations for the 4 nm YFeO$_3$ film was determined by the adjustment of (0 0 l) reflections from the orthorhombic YFeO$_3$ (**d**) and (h 0 0) reflections from orthorhombic YFeO$_3$ (**e**).

## 5. Summary

We investigated the ultrathin yttrium orthoferrite films prepared by magnetron sputtering by the Mössbauer reflectivity method, XRD, and GI-XRD. It has been discovered that depending on the thickness, our films have a complicated crystallographic structure, though the predominating phase is orthorhombic yttrium orthoferrite, but they contain the inclusions of the YIG phase as well as the hexagonal YFeO$_3$ phase. It has been observed that the orientation of the AFM axes of YFeO$_3$ depends on the film thickness, namely, for the ultrathin film (<~10 nm), the AF axes in YFeO$_3$ are not in the surface plane.

From the general considerations, one can argue that the presence of very small (less than 1% wt.) amounts of YIG inclusions may not significantly influence the magnetic properties of the films in aspects such as the formation of the magnetic ordering and intra-layer spin-dependent phenomena. For the series of films investigated, substantial variations of the relative content of the YIG phase were observed. Their strong dependence on the film thicknesses and details of the technological process of film preparation was revealed by the advanced methods used.

The observed peculiarities of the crystal and magnetic properties of the ultrathin YFeO$_3$ films are important for their potential applications in antiferromagnetic spintronics and magnetic materials for informatics.

**Supplementary Materials:** The following supporting information can be downloaded at: https://www.mdpi.com/article/10.3390/magnetism2040023/s1, Table S1. XRD patterns and numerical data for the orthorhombic YFeO$_3$; Table S2. for the hexagonal YFeO$_3$; Table S3. for the Al$_2$O$_3$ (Sapphire); Figure S1: 100 nm layer of (001) orthorhombic YFeO$_3$ on *r*-Al$_2$O$_3$; Figure S2: 100 nm layer of (010) orthorhombic YFeO$_3$ on *r*-Al$_2$O$_3$; Figure S3: 100 nm layer of hexagonal (0 0 0 1) YFeO$_3$ on *r*-Al$_2$O$_3$.

**Author Contributions:** Conceptualization—A.N.; Methodology—A.N., M.A. and E.P.; Software—M.A. and I.S.; Validation—I.S. and M.A.; Formal analysis—A.N., I.S., R.B. and M.A.; Investigation—A.N., V.I., I.G., I.S. and O.K.; Resources—A.N. and E.P.; Data curation—I.G., V.I., R.B., O.K. and

M.A.; Writing, original draft preparation—M.A.; Writing, review and editing—E.P., A.N., R.B. and M.A.; Visualization—I.S. and M.A.; Supervision—E.P. and A.N.; Project administration—A.N.; Funding acquisition—A.N. and E.P. All authors have read and agreed to the published version of the manuscript.

**Funding:** The X-ray structure investigation was supported in part by the Ministry of Science and Higher Education of the Russian Federation (No. 075-15-2021-1350, internal number 15.SIN.21.0004).

**Acknowledgments:** The authors are grateful to the ESRF administration and personally to A.I. Chumakov for performing the firsthand measurements at ID18 (HC4300 proposal) during COVID restrictions when we were able to participate in the experiment only online. The research was carried out within the state assignment of the Ministry of Science and Higher Education of the Russian Federation (theme "Function" No. 122021000035-6).

**Conflicts of Interest:** The authors declare no conflict of interest.

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
