# Peer review of "Mössbauer Synchrotron and X-ray Studies of Ultrathin YFeO3 Films"

_2673-8724, doi:10.3390/magnetism2040023_

Round 1

Reviewer 1 Report

Andreeva et al. present a study characterizing ultrathin films of YFeO3 through x-ray diffraction and Mossbauer spectroscopy. As the authors point out, materials in this geometry are highly relevant in device applications. I recommend this study for publications subject to the following points being addressed:

1.      English writing must be improved. There are a number of instances with plural-singular errors, comma errors, sentence structure errors, and use of language which is not appropriate for publication in a scientific journal. For example, unusual instructions to the reader are given twice within the same paragraph: lines 139 – 140: “pay your attention to…” and line 149: “remember that the radiation…”.

2.      The authors mention the AFM axis but do not clearly define what this means. Perhaps adding a schematic to figure 4 would be helpful to illustrate this.

3.      The authors are not surprised by the occurrence of YIG in their thin films and cite references about the stability of each phase. However, the purpose of the study was analysis of ultrathin films due to their usefulness in modern technologies. The authors should discuss how the unavoidable YIG content is expected to impact applications such as AFM spintronics.

Author Response

We are grateful to all the reviewers for their work in carefully analyzing our article and for the important comments that we all tried to take into account in the right way.

Reply to Referee 1

  1. English writing must be improved. There are a number of instances with plural-singular errors, comma errors, sentence structure errors, and use of language which is not appropriate for publication in a scientific journal. For example, unusual instructions to the reader are given twice within the same paragraph: lines 139 – 140: “pay your attention to…” and line 149: “remember that the radiation…”.

We carefully checked the text and inserted several corrections. If it is still not good we’ll apply for the suggested editing services.

The sentence

“(pay your attention to the change of the multiplet splittings in Figure 2b)”

is changed to:

“The most essential feature is the change of the multiplet splittings.”

The sentence

“(remember that the radiation from SMS is p-polarized, polarization dependences of the line ratio in Mössbauer sextet for this case is analyzed in e.g. [39])

is changed to:

“Note that for the interpretation of the line ratio in magnetic sextets it is important to take into account the p-polarization of radiation from SMS. Such analysis was presented for example in [39].”

  1. The authors mention the AFM axis but do not clearly define what this means. Perhaps adding a schematic to figure 4 would be helpful to illustrate this.

The term “AFM axis” is commented by “(the direction along which the AFM ordered moments align)”. In order to explain the angles used in (1) we added the picture (Figure 5), because the additional designations in Fig.4 make it too complicated.

  1. The authors are not surprised by the occurrence of YIG in their thin films and cite references about the stability of each phase. However, the purpose of the study was analysis of ultrathin films due to their usefulness in modern technologies. The authors should discuss how the unavoidable YIG content is expected to impact applications such as AFM spintronics.

We added the following paragraph to the “Summary”

From general considerations, one can argue that, the presence of very small (less than 1% wt.) amounts of the YIG inclusions may not significantly influence the magnetic properties of the films in such aspects as formation of the magnetic ordering and intra-layer spin-dependent phenomena. For the series of films investigated, substantial variations of the relative content of YIG phase were observed. Their strong dependence on film thicknesses and details of technological process of film preparation was revealed by the used advanced methods.

Reviewer 2 Report

In this study, the authors have prepared ultrathin YFeO3 by magnetron sputtering on rAl2O3 substrates. Mössbauer synchrootron and xrd properties were studied. The work is presented nicely, and this manuscript can be accepted. below are my comments

authors should include the reference x-ray patterns of the substrate as well as YFeO3 for confirmation of the corresponding x-ray peaks.

Author Response

We are grateful to all the reviewers for their work in carefully analyzing our article and for the important comments that we all tried to take them into account in the right way.

Reply to Referee 2

authors should include the reference x-ray patterns of the substrate as well as YFeO3 for confirmation of the corresponding x-ray peaks.

In our work we did not measure XRD from the substrate and bulk YFeO3. In the literature the X-ray powder diffraction has been measured for the corresponded polycrystalline samples. These patterns include the huge numbers of peaks which exact angular positions is not clearly seen. Therefore, we decided to add the Supplementary material where the theoretical XRD for these compounds are given as pictures and Tables (based on the known CIF files). The reflections for our almost monocrystalline films in the chosen geometries are marked in the Tables.

Reviewer 3 Report

Manuscript number: magnetism-1817264 Magnetism MDPI (type of the paper: Article)

TITLE: Mössbauer Synchrotron and X-ray Studies of Ultrathin YFeO3 2 Films

AUTHORS: Marina Andreeva, Roman Baulin, Aleksandr Nosov, Igor Gribov, Vladimir Izyurov, Oleg Kondratev, Ilia Subbotin, Elkhan Pashaev

The first review of the manuscript

Overall description of the manuscript

In the manuscript entitled “Mössbauer Synchrotron and X-ray Studies of Ultrathin YFeO3 2 Films” authored by Marina Andreeva, Roman Baulin, Aleksandr Nosov, Igor Gribov, Vladimir Izyurov, Oleg Kondratev, Ilia Subbotin, Elkhan Pashaev, the authors present experimental investigations of properties of experimental studies of ultrathin YFeO3 films with using several complementary methods: (i) Mossbauer reflectivity method and (ii) X-ray diffraction studies. The results reveal some interesting features of investigated materials, e.g., for thin films with different thicknesses spin reorientation was detected

The paper fits the journal scope. The English language in the manuscript is good. The paper has 12 pages and includes 50 references (equivalent to 2 pages), and 7 figures (about 2 and 1/2 pages) – effectively about 7 pages of the main text. The diagrams (figures) are clear, they are essential and their captions are informative. The title clearly and concisely conveys the topic of the article. The abstract quite well describes the content of the manuscript. The findings look correctly. The discussion and conclusions are supported by the results.

In my opinion, the manuscript can be published in the present form. I strongly believe this paper is suitable for publication in “Magnetism” MDPI journal as an regular article. The topic of the paper, which is strongly associated with interesting antiferromagnetic materials, attracts a lot of attention (because of their potential applications in sensors and magnetic devices) can be interesting for some groups of scientists.

Author Response

Thank you for the positive evaluation of our manuscript.

There are no comments requiring the corrections.

Reviewer 4 Report

Referee report 

magnetism-1817264-peer-review-v1

Mössbauer Synchrotron and X-ray Studies of Ultrathin YFeO3 Films

Marina Andreeva et al.

This manuscript discusses details of yttrium orthoferrite, YFeO3, which is a very interesting material for antiferromagnetic 

spintronics. And especially, the properties of thin film materials are outstanding.

Thus, the topic is well suited for Magnetism.

The present manuscript comprises 7 figures, no table, and 50 references are given, which provide a good overview on this field.

The manuscript is well organized, well arranged and well written. The English is quite good; only minor changes are necessary like 

"angels".

All figures are well prepared and the lettering is properly sized.

The manuscript provides all necessary experimental details, and the discussion is performed well and thorougly considering all

details.

Bedides the already mentioned points concerning the English, there are only some minor points to be considered:

# Please take care for spaces throughout the manuscript, in formulae, in equations, etc.

# Define rAl2O3 substrates in the abstract.

# mbar is not a SI unit.

# line 164ff: the degree sign does not correspond to the stanard one °.

# Please write proper formulae also in the reference titles. 

Overall, this manuscript contains interesting information. Thus, the manuscript may be accepted for publication provided that the little problems are treated well.

Author Response

We are grateful to all the reviewers for their work in carefully analyzing our article and for the important comments that we all tried to take them into account in the right way.

Reply to Referee 4

"angels".     corrected

# Please take care for spaces throughout the manuscript, in formulae, in equations, etc.

Sorry, we do not know how to arrange the line spacing where some formula are included

# Define rAl2O3 substrates in the abstract.

Corrected to “r -plane ()-oriented Al2O3 substrates (r-Al2O3)”

# mbar is not a SI unit.

We added in Pa   (9×10-3 mbar=0,9 Pa)

# line 164ff: the degree sign does not correspond to the standard one °.

Corrected everywhere

# Please write proper formulae also in the reference titles. 

  Corrected
